# Orientin Improves Substrate Utilization and the Expression of Major Genes Involved in Insulin Signaling and Energy Regulation in Cultured Insulin-Resistant Liver Cells

**DOI:** 10.3390/molecules26206154

**Published:** 2021-10-12

**Authors:** Sithandiwe E. Mazibuko-Mbeje, Sinenhlanhla X. H. Mthembu, Andani Tshiitamune, Ndivhuwo Muvhulawa, Fikile T. Mthiyane, Khanyisani Ziqubu, Christo J. F. Muller, Phiwayinkosi V. Dludla

**Affiliations:** 1Department of Biochemistry, Mafikeng Campus, North-West University, Mmabatho 2735, South Africa; sinenhlanhla.mthembu@mrc.ac.za (S.X.H.M.); anditshiita@gmail.com (A.T.); mn.muvhulawa@gmail.com (N.M.); fikilemthiyane4@gmail.com (F.T.M.); ziqubukhanyisani@gmail.com (K.Z.); 2Biomedical Research and Innovation Platform, South African Medical Research Council, Tygerberg 7505, South Africa; christo.muller@mrc.ac.za (C.J.F.M.); phiwayinkosi.dludla@mrc.ac.za (P.V.D.); 3Department of Biochemistry and Microbiology, University of Zululand, KwaDlangezwa 3886, South Africa; 4Division of Medical Physiology, Stellenbosch University, Tygerberg 7505, South Africa

**Keywords:** orientin, type 2 diabetes, dyslipidemia, insulin resistance, energy metabolism, palmitate

## Abstract

Our group has progressively reported on the impact of bioactive compounds found in rooibos (*Aspalathus linearis*) and their capacity to modulate glucose homeostasis to improve metabolic function in experimental models of type 2 diabetes. In the current study, we investigated how the dietary flavone, orientin, modulates the essential genes involved in energy regulation to enhance substrate metabolism. We used a well-established hepatic insulin resistance model of exposing C3A liver cells to a high concentration of palmitate (0.75 mM) for 16 hrs. These insulin-resistant liver cells were treated with orientin (10 µM) for 3 h to assess the therapeutic effect of orientin. In addition to assessing the rate of metabolic activity, end point measurements assessed include the uptake or utilization of glucose and palmitate, as well as the expression of genes involved in insulin signaling and regulating cellular energy homeostasis. Our results showed that orientin effectively improved metabolic activity, mainly by maintaining substrate utilization which was marked by enhanced glucose and palmitate uptake by liver cells subjected to insulin resistance. Interestingly, these effects can be explained by the improvement in the expression of genes involved in glucose transport (*Glut2*), insulin signaling (*Irs1* and *Pi3k*), and energy regulation (*Ampk* and *Cpt1*). These preliminary findings lay an important foundation for future research to determine the bioactive properties of orientin against dyslipidemia or insulin resistance in reliable and well-established models of type 2 diabetes.

## 1. Introduction

According to global health surveillance organizations, such as the World Health Organization (WHO) [1] and International Diabetes Federation (IDF) [2], metabolic diseases such as diabetes mellitus consistently rank among the leading causes of death each year, especially over the last decade. Apart from contributing significantly to reduced life expectancy, diabetes, particularly its predominant form type 2 diabetes (T2D), is known to induce detrimental metabolic abnormalities that are consistent with poor-quality of living of those affected [3,4]. Notably, the hallmarks of T2D, such as persistently raised blood glucose levels (a state of hyperglycemia); insulin resistance; and aberrant lipid profiles, including elevated cholesterol concentrations (a state of dyslipidemia), are implicated in the development and aggravation of certain chronic metabolic conditions, such as muscle degeneration, myocardial dysfunction, and liver toxicity [2,5,6]. As a result, there is an increasing interest in understanding the pathological mechanisms related to the development of T2D to uncover mechanisms to find therapeutics to improve metabolic function and reverse the detrimental effects linked with hyperglycemia or dyslipidemia. 

Briefly, overloading cellular compartments with saturated fats, such as palmitate, has been a popular and reliable model to study the lipotoxicity effects of dyslipidemia in vitro. In agreement, our research laboratory has consistently applied this in vitro method for the early screening of various bioactive active compounds for their therapeutic effects against metabolic abnormalities [7,8]. In fact, due to its envisaged health benefits against metabolic disease, the major focus has been placed on unraveling the therapeutic mechanisms of bioactive compounds found in rooibos (*Aspalathus linearis*) against T2D-related complications [9]. Consistently, in addition to the phenylpyruvic acid-2-*O*-*β*-D-glucoside, which is known to possess some antidiabetic properties in vitro [9], rooibos is known to contain high levels of *C*-glucosyl flavonoids, especially the dihydrochalcones, aspalathin and nothofagin, which are increasingly explored for their ameliorative effects against diverse metabolic complications [9,10]. Rooibos also contains high levels of dietary flavones, including orientin, which are known to contain abundant antioxidant properties that could play an essential role in blocking the detrimental effect of lipid overload or dyslipidemia [9,10]. Available literature, predominantly from our group, has reported on the impact of these rooibos bioactive compounds in modulating prime mechanisms involved in the development of insulin resistance and T2D, such as the regulation of phosphoinositide 3-kinase and protein kinase B (Pi3k/Akt) as well as the activation of AMP-activated protein kinase (AMPK) to improve cellular metabolism, through the use of experimental models of metabolic disease [7,9,11,12].

Importantly, there is already evidence indicating that dietary flavones, particularly the *C*-glycosyl flavone isomers, orientin and isoorientin, can improve glucose metabolism in cultured adipocytes [13,14], or attenuate vascular inflammation in human umbilical vein endothelial cells and mice [15], and display other biological properties [16,17]. However, to date, there is lack of data reporting on the impact of this dietary flavone on metabolic dysregulations within the liver, especially those related with insulin resistance. It is extremely important to determine, considering that the liver remains a major therapeutic target to improve substrate metabolism and energy regulation [18,19]. Thus, this research communication provides essential preliminary findings that inform how the dietary flavone, orientin, modulates the essential genes involved in energy regulation to enhance improve substrate metabolism (the uptake of glucose and palmitate) and enhance ATP (adenosine triphosphate) production, in an experimental model of hepatic insulin resistance.

## 2. Results

### 2.1. Effect of Orientin on Metabolic Activity in Palmitate-Exposed Liver Cells

Metabolic assays, such as MTT (3-(4,5-dimethylthiazol-2-yl)-2,5-diphenyltetrazolium bromide), and ATP assays are widely used to measure the metabolic activity of cells in response to different types of treatments after induction of insulin resistance. Here, exposure of hepatic cells in elevated concentrations of palmitate resulted in a significant decrease in both MTT activity (*p* < 0.05) and ATP production (*p* < 0.01) (Figure 1A,B). However, treatment with orientin, either as a monotherapy or in combination with insulin, increased both MTT activity (*p* < 0.001) and ATP production (*p* < 0.001) (Figure 1A,B).

### 2.2. Effect of Orientin on Substrate Metabolism in Palmitate-Exposed Liver Cells

The uptake of major energy-generating substrates, such as glucose and palmitate, provides an important estimation of glucose and lipid homeostasis in response to metabolic stress, including insulin resistance. Here, exposing hepatic cells to an elevated concentration of palmitate resulted in a significant reduction in the uptake of glucose (*p* < 0.05) and palmitate (*p* < 0.05) (Figure 2A,B). However, treatment with orientin as a monotherapy was effective in improving glucose (*p* < 0.05) and palmitate uptake (*p* < 0.001) (Figure 2A,B). Notably, data reporting on the combination effect of orientin and insulin on substrate regulation remain inconclusive.

### 2.3. Effect of Orientin in Modulating the Expression of Different Genes Involved in Insulin Signaling in Palmitate-Exposed Liver Cells

Hepatic glucose transportation via the basal regulator, glucose transporter 2 (*Glut2*), and the modulation of insulin signaling through the insulin receptor substrate 1 (*Irs1*)/ phosphatidylinositol 3 kinase (*Pi3k*) pathway, are well-accomplished mechanisms involved in glucose and lipid homeostasis within the liver. Here, exposing hepatic cells to an elevated concentration (0.75mM) of palmitate resulted in significantly reduced mRNA levels of *Irs1* (*p* < 0.001), *Pi3k* (*p* < 0.01), and *Glut2* (*p* < 0.01) in cultured hepatic cells (Figure 3A–C). This was reversed by orientin treatment as a monotherapy (*p* < 0.001, *p* < 0.5, *p* < 0.5, respectively) (Figure 3A–C).

### 2.4. Orientin Modulate Energy Metabolism in Palmitate-Induced Insulin-Resistant C3A Liver Cells

The current study also showed that exposure of hepatic cells to an elevated concentration of palmitate effectively modulated the mRNA expression of genes responsible for energy regulation and mitochondrial beta-oxidation, such as *Ampk* and carnitine palmitoyl transferase 1 (*Cpt1*) (significantly decreased at *p* < 0.001) (Figure 4A,B). Interestingly, orientin treatment as a monotherapy (*p* < 0.01 and *p* < 0.001, respectively) markedly increased the expression of these genes in insulin-resistant hepatic cells (Figure 4A,B). Notably, data further indicate that the combination effect of orientin and insulin might affect these genes; however, this does not show much difference when compared to orientin monotherapy or to treatment with insulin (as a positive control).

## 3. Discussion and Conclusions

The liver remains one of the essential organs that has long been targeted to investigate the pathological mechanisms of insulin resistance, largely because of its prime role in controlling glucose homeostasis [18,19]. This is consistent with the classical features of T2D which represent abnormal insulin signaling paralleled by severely reduced hepatic glucose production and enhanced liver lipid synthesis, i.e., consequences that aggravate the state of hyperglycemia and hyperlipidemia [18,19]. This also explains increased interest in targeting the amelioration of hepatic insulin resistance using novel bioactive compounds to revert T2D-related complications [7,18,19]. As such, the current study reports on the impact of the bioactive compound, orientin, on substrate utilization, including the modulation of genes implicated in energy regulation, using an established model of hepatic insulin resistance. Indeed, as previously reported, exposing cultured liver cells to high concentrations of palmitate could effectively induce a state of hepatic insulin resistance. This was characterized by the significant diminished uptake of glucose or palmitate that occurred concurrently to reduced levels of cell metabolic activity or energy (ATP) production, suggesting that mitochondrial beta-oxidation was inhibited. Further, indicating a compromised state of metabolic function, liver cells exposed to palmitate displayed an altered glucose transport and insulin signaling as demonstrated by suppressed expression of genes, such as *Glut2*, *Irs1*, and *Pi3k*, which are essential for effective glucose transport and insulin-dependent glucose utilization. 

Alternatively, it was clear that treatment with orientin was able to reverse palmitate-induced hepatic insulin resistance in part by improving substrate utilization, for example, it enhanced glucose uptake and effectively modulated the expression of genes coding for glucose transport (*Glut2*) and insulin signaling (*Irs1* and *Pi3k*) (Figure 5). Interestingly, well-established therapies that are used to manage T2D, such as physical exercise, can also target this pathway to improve cellular metabolism in conditions of metabolic stress [20]. Nonetheless, consistent with other bioactive compounds found in rooibos, such as aspalathin, orientin displays a strong potential to enhance the mRNA expression levels of *Ampk* and *Cpt1*, which are all part of an essential mechanism involved in insulin-independent glucose regulation. In addition, effective upregulation of *Ampk* and *Cpt1* could suggest that orientin plays a major role in controlling mitochondrial beta-oxidation and energy metabolism. This is consistent with in vitro biological effects observed with using another dietary flavone (isoorientin) in cultured adipocytes, also exposed to high levels of palmitate [12,21]. These are all vital molecular mechanisms that are linked with the therapeutic actions of well-known drugs such as metformin, which are accredited for targeting the reversal of hepatic insulin resistance to improve metabolic function [22]. In agreement, the current communication certainly affirms that orientin improves the substrate utilization within the liver subjected to insulin resistance, in part by regulating essential mechanisms of insulin signaling and energy metabolism. 

Currently, the use of antidiabetic agents, such as insulin and metformin, has certainly prolonged the lives of patients with diabetes [23]; however, their long-term effects are in question due to the continued rise in T2D-related mortalities [3,4]. Thus, beyond understanding the pathogenesis of diabetes, there is an urgent need to establish alternative effective therapeutic drugs that can be used, either as a monotherapy or in combination with current antidiabetic agents, to manage diabetes-related complications. There is already accumulative curiosity in using nutraceuticals or natural-derived compounds as supplements, as a monotherapy or in combination with hypoglycemic drugs, to improve basic metabolic function [11,24,25]. This also translates to everyday living of consuming beverages such as herbal teas, including rooibos, which are known to contain abundant bioactive compounds, with potential therapeutic properties to affect metabolic function [9,10]. In agreement, a large body of literature suggests that many food-derived bioactive compounds, such as aspalathin, curcumin, and resveratrol, can positively affect glucose or lipid metabolism, in part by controlling major cellular response mechanisms such as activation of insulin dependent or independent signaling pathways in conditions of metabolic stress [11,24,25]. In fact, there is a need to develop bioactive compounds found in foods or beverages to useful nutraceuticals that could be effectively used to manage metabolic diseases, such as T2D [11]. This is supported by growing preclinical evidence indicating that potential nutraceutical compounds, including dietary flavones found in daily consumed foods such as fruits and vegetables, as well as beverages such as tea, can modulate diverse biochemical reactions that are essential for improving metabolic function [26,27,28,29]. However, although such bioactive properties are acknowledged, certain critical issues, such as the determining the quantity of these active compounds in foods, as well as establishing their oral bioavailability, need to be resolved to better understand the therapeutic effects.

Collectively, this study lays an important foundation for future investigations to confirm these findings in other well-established experimental models of insulin resistance or T2D. This is in line with also checking the comparative or combination use of this flavone with other antidiabetic agents such as metformin, since there were no synergistic effects observed in combining orientin and insulin to reverse hepatic insulin resistance in the current study. 

## 4. Materials and Methods

### 4.1. Reagents and Kits Used

Human C3A liver cells (ATTC Cat. No. CRL-10741) were purchased from the American Type Culture Collection (Manassas, VA, USA). Orientin (≥ 97.0% purity), MTT, and dimethyl sulfoxide (DMSO) were obtained from Sigma-Aldrich (St Louis, MO, USA), while 2-Deoxy-[^3^H]-D-glucose and ^14^C palmitate were obtained from American Radiolabeled Chemicals (St Louis, MO, USA). Eagle’s minimum essential medium (EMEM), Dulbecco’s phosphate buffered saline (DPBS, pH 7.4 with calcium and magnesium), penicillin-streptomycin, and ViaLight plus ATP kits were from Lonza (Basel, Switzerland). PCR probes, beta-2 microglobulin (*B2m*), *Pi3k*, *Irs1*, *Glut2*, *Cpt1*, and *Ampk* were purchased from Thermofisher (Waltham Massachusetts, USA). All other cell culture reagents were obtained from Sigma-Aldrich (St Louis, MO, USA).

### 4.2. Cell Culture, Establishing the Experimental Model of Insulin Resistance, and Orientin Treatment

In this study, C3A hepatocytes, acquired from American Type Culture Collection (Manassas, Virginia, USA) were utilized for an in vitro study of hepatic insulin resistance. Briefly, C3A liver cells were cultured and treated, as previously described by Mazibuko-Mbeje et al. (2019) [7]. Upon confluency, insulin resistance was induced in liver cells by exposing C3A liver cells to palmitate (0.75 mM), which was conjugated with 2% bovine serum albumin (BSA), in EMEM supplemented with 8 mM of glucose, while normal controls were cultured in the absence of palmitate for 16 hr. After induction of insulin resistance, orientin stock solution was prepared in DMSO, with the final concentration of 10 μM prepared in EMEM containing 8 mM of glucose. Thereafter, relevant assays were conducted. Notably, the cytotoxicity was avoided in cell culture treatments, as the final concentration of DMSO within the final working concertation of orientin working solution was < 0.001%, consistent with previously published cell culture considerations [30]. The dose and treatment duration with the compound of interest (orientin) was based on an already established protocol [31].

### 4.3. Determining Metabolic Activity

To investigate the effect of orientin on metabolic activity, a well-known protocol of Mosmann (1983) used was slightly modified [32]. Briefly, after treatment with orientin for 3 h, MTT solution was added, as described by a previously published protocol [30]. Furthermore, ATP was determined using the ViaLight plus ATP kit. The assay was conducted following manufacturer’s recommendations. Thereafter, both MTT and ATP absorbance and luminescence were read using a BioTek ELx800/FLx800 plate reader and Gen 5 software for data acquisition (BioTek Instruments Inc., Winooski, VT, USA).

### 4.4. Assessing Substrate Metabolism: Glucose and Palmitate Uptake

To assess substrate metabolism, radioactive glucose (^3^H-2-DOG) and ^14^C palmitate uptake assays were conducted, as previously described by Mazibuko et al. (2019) [7]. Briefly, cells were labelled with 0.5 µCi/mL of 3H-2-DOG for glucose uptake or 1 µCi/mL ^14^C for palmitate uptake in medium supplemented with or without orientin for 3 h at 37 °C in 5% CO2 and humidified air for determination of glucose and palmitate uptake, respectively. Insulin (1 µM) was added during the last 15 min in each experiment as a positive control, consistent with our published study [7]. Radiolabeling was terminated by washing cells in PBS. Thereafter, cells were lysed with 0.1 M of NaOH before ^3^H-2-DOG or ^14^C activity, and the lysate was assessed by liquid scintillation (2220 CA, Packard Tri-Carb series, PerkinElmer, Downers Crove, IL, USA).

### 4.5. mRNA Expression Analysis Using RT-PCR

Gene expression was analyzed using a method previously described and published by Mazibuko et al. (2021) [33]. Briefly, total RNA was extracted from C3A liver cells using QIAzol lysis reagent, and was cleaned and reverse transcribed into complementary DNA (cDNA) using QuantiTect Reverse Transcription kit (Qiagen, Hilden, German), according to the manufacturer’s instructions. TaqMan gene expression assays used are included in Table 1. The quantitative RT-PCR conditions were as follows: 95 °C for 10 min, followed by 40 cycles of 95 °C for 15 s and 60 °C for 1 min in QuantStudio™ 7 Flex Real-Time PCR System (Thermo Scientific TM, MA, USA). Gene expression data were normalized to *B2m*.

## Figures and Tables

**Figure 1 molecules-26-06154-f001:**
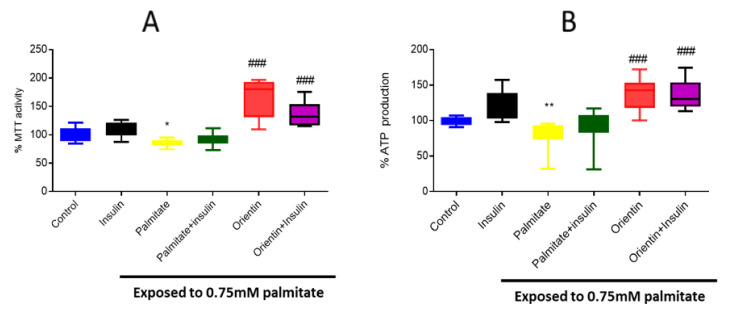
Effect of orientin on metabolic activity, measured by MTT activity (**A**) and ATP production (**B**) in C3A liver cells exposed to high (0.75 mM) palmitate concentration. Briefly, insulin-resistance was induced with 0.75 mM of palmitate for 16 h in C3A liver cells, followed by treatment with or without 10 µM of orientin for 3 h and 1 µM of insulin, which was added in the last 15 mins. Thereafter, cell metabolic activity was estimated utilizing MTT and ATP assays. Results are expressed as mean ± SD of 3 independent experiments. The bars depict statistical differences at * *p* < 0.05, ** *p* < 0.01, versus the experimental control set at 100% and ^###^
*p* < 0.001 versus palmitate control.

**Figure 2 molecules-26-06154-f002:**
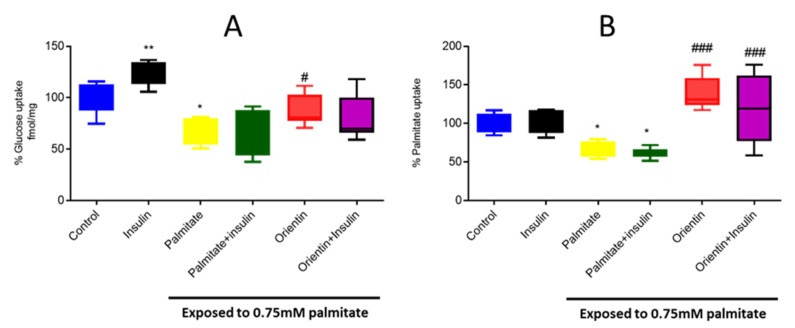
Effect of orientin on substrate utilization as measured by uptake of glucose (**A**) and palmitate (**B**) in C3A liver cells exposed to high (0.75 mM) palmitate concentration. Briefly, insulin-resistance was induced with 0.75 mM of palmitate for 16 h in C3A liver cells, followed by treatment with or without 10 µM of orientin for 3 h and 1 µM of insulin, which was added in the last 15 min. Thereafter, glucose uptake was measured using [3H]-2-deoxy-D-glucose and palmitate uptake was determined using 14C palmitate. Results are expressed as mean ± SD of 3 independent experiments. The bars depict statistical differences at * *p* < 0.05, ** *p* < 0.01, versus the experimental control set at 100% and ^#^
*p* < 0.05 and ^###^
*p* < 0.001 versus the palmitate control.

**Figure 3 molecules-26-06154-f003:**
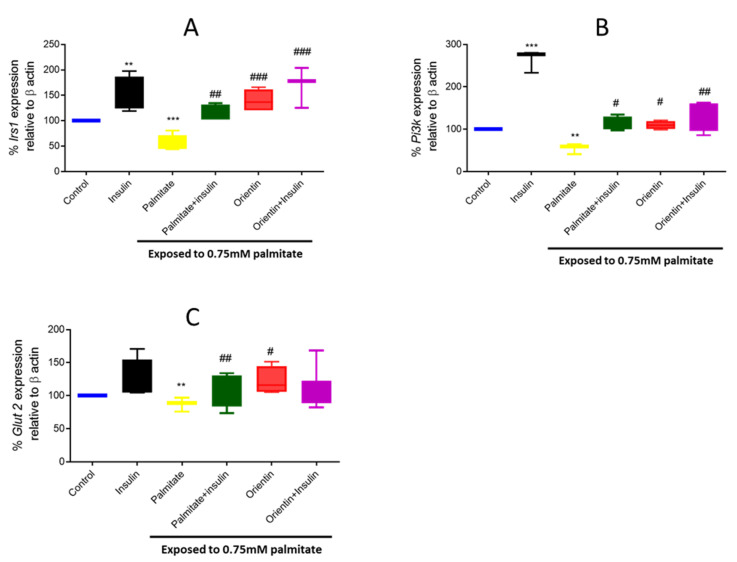
Effect of orientin on the expression of different genes involved in insulin signaling such as insulin receptor substrate 1 (*Irs1*) (**A**), phosphatidylinositol 3 kinase (*Pi3K*) (**B**) and glucose transporter 2 (*Glut2*) (**C**) in C3A liver cells exposed to high (0.75 mM) palmitate concentrations. Briefly, insulin resistance was induced with 0.75 mM of palmitate for 16 h in C3A liver cells, followed by treatment with or without 10 µM of orientin for 3 h and 1 µM of insulin. Thereafter, gene expression was investigated using qPCR. Results are expressed as mean ± SD of 3 independent experiments. The bars depict statistical differences at ** *p* < 0.01, *** *p* < 0.001, versus the experimental control set at 100% and ^#^
*p* < 0.05, ^##^
*p* < 0.01, ^###^
*p* < 0.001 versus palmitate control.

**Figure 4 molecules-26-06154-f004:**
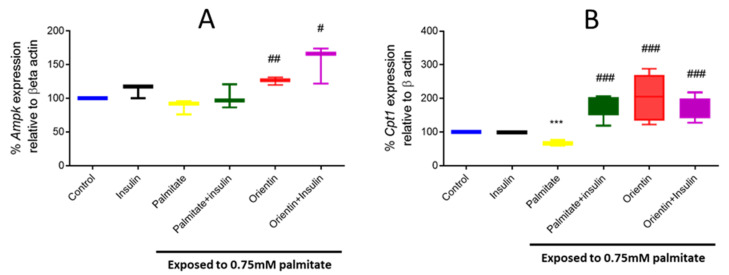
Effect of orientin on the expression of genes involved in energy metabolism, such as 5′adenosine monophosphate-activated protein kinase (*Ampk*) (**A**) and carnitine palmitoyl transferase 1 (*Cpt1*) (**B**) in C3A liver cells exposed to high (0.75 mM) palmitate concentration. Briefly, insulin resistance was induced with 0.75 mM of palmitate for 16 h in C3A liver cells, followed by treatment with or without 10 µM of orientin for 3 hrs and 1 µM of insulin. This was then followed by investigating genes in energy metabolism using qPCR. Results are expressed as mean ± SD of 3 independent experiments. The bars depict statistical differences at *** *p* < 0.001, versus the experimental control set at 100% and ^#^
*p* < 0.05, ^##^
*p* < 0.01, ^###^
*p* < 0.001 versus palmitate control.

**Figure 5 molecules-26-06154-f005:**
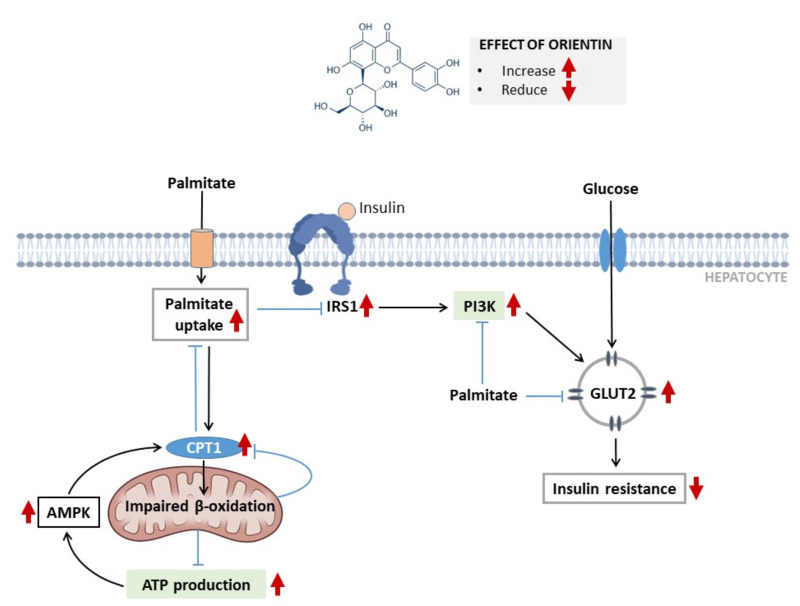
The proposed molecular mechanism by which orientin modulates substrate metabolism to ameliorates palmitate-induced insulin resistance in liver cells. Briefly, exposing liver cells to high palmitate doses resulted in impaired beta-oxidation, in relation to markedly reduced expression of carnitine palmitoyl transferase 1 (*Cpt1*) and palmitate uptake, which in turn translated to decreased ATP production. On the other hand, palmitate inhibited insulin signaling genes, such as insulin receptor substrate 1 (*Irs1*), phosphatidylinositol 3 kinase (*Pi3k*), and glucose transporter 2 (*Glut2*), which was consistent with insulin resistance. However, orientin treatment improved beta-oxidation as evidenced by increased AMP-activated protein kinase (*Ampk*) and (*Cpt1*), accompanied by increased palmitate uptake and ATP production. Moreover, orientin increased the expression of *Irs1*, *Pi3k*, and *Glut2*, and increased glucose uptake to potentially alleviate insulin resistance.

**Table 1 molecules-26-06154-t001:** List of gene expression probes used in this study.

Gene Probes	Assay ID
Beta-2 Microglobulin (*B2m)*	Hs99999907-m1
Phosphatidylinositol 3 Kinase (*Pi3k*)	Hs00933163-R1
Insulin receptor substrate 1 (*Irs1*)	Hs00178563-R1
Glucose transporter 2 (*Glut2*)	Hs00165775-R1
Carnitine Palmitoyl transferase 1 (*Cpt1*)	Hs00912671-m1
5’adenosine monophosphate activated protein kinase (*Ampk*)	Hs00178903-m1

## Data Availability

Data related to search strategy, study selection and extraction items will be made available upon request after the manuscript is published.

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
