# Peer review of "Orientin Improves Substrate Utilization and the Expression of Major Genes Involved in Insulin Signaling and Energy Regulation in Cultured Insulin-Resistant Liver Cells"

_molecules, 2021, doi:10.3390/molecules26206154_

Round 1

Reviewer 1 Report

The manuscript entitled “Orientin improves substrate utilization and the expression of major genes involved in insulin signaling and energy regulation in cultured insulin resistant liver cells” is a wonderful study and presented in a complete and clear manner. It is also conceded as extension for previous studies illustrate other effects for this flavone compound

Author Response

We appreciate the positive response and the time made available by the reviewer to assess our manuscript.

Reviewer 2 Report

 There should be a known drug or standard drug like metformin in all experiments. A comparison study between standard drug and orientin would be better.

-          Figure 1. MTT activity and ATP production. There is no significant difference between orientin and orientin + insulin. Why?

-          Figure 2. effect of orientin on substrate metabolism. “However, treatment with orientin, either as a monotherapy or in combination with insulin, was effective in improving glucose (p<0.05) and pal-153 mitate uptake (p<0.001) (Figure 2A, B).” The broad statement has flaws. Figure 2A there is no significance difference between orientin and orientin + glucose, so saying combination of orientin and insulin improve both is not correct. Figure 2B orientin and insulin bar is covering a very broad range of percentage of palmitate uptake compared to others, I think we can not make any judgment by this figure.

-          Figure 3B. Looking at the figure it seems that orientin + insulin therapy is bad compare to insulin alone and same compare to orientin. Why?

-          Figure 4B. There is no difference between orientin and orientin +insulin again. So a broad statement saying combination of orientin and insulin improving the gene expression does not fit.

-          Discussion part. I think this part is more like an introduction not a discussion or experiment done by the author. It will be better if Figure 5 can be included in introduction not in discussion because this is already known. 

Reviewer 3 Report

The submitted paper describes the influence of the dietary flavone, orientin, on the essential genes involved in insulin signaling and energy regulation in cultured insulin resistant liver cells. In my opinion the study is valuable and the presented results are convincing. The paper is concise and well written.

There are only two minor points which should be addressed by the authors:

- line 91: double ‘The’,

- line 92: ‘upon’ vs. ‘uupon’.

Author Response

We appreciate the positive response and the time made available by the reviewer to assess our manuscript. The errors identified by the reviewer have been corrected, and we value your input in improving the quality of our manuscript.

Reviewer 4 Report

In this communicaition, Mazibuko-Mbeje and colleagues show the biological activity of natural flavone Orientin in the recovery of metabolic function in insulin-resistant hepatic cells. The study, while very small and limited in scope, shows valid methodology and promising results.

Some minor points regarding this manuscript:

Introduction
The authors go from a general description of T2D to a description of in vitro T2D models quite abruptly (lines 49-50). For the sake of clarity, it would probably be best to describe the bioactive compounds found in rooibos, and then the methods used to assay their activity.

Materials and methods
l. 92:        there is a typo ("uupon")
ll. 96-98:    the authors should specify the concentration of the orientin stock solution or at least the % of DMSO in the final 10μM solution in EMEM.
ll. 103-105:    it is not clear what the authors mean with "as described in us in-in house published protocol 104 [20]." I assume it is a previously published protocol. It would probably better to rephrase this sentence for better readability.

Results
l. 151:        there is a typo ("ofof")
Figure 3:    the authors should enlarge this figure - it is too hard to read.

Discussion and conclusions
ll.210-214:    the authors must mention some critical points regarding nutraceutical compounds in food and beverages, such as the quantity of the active compounds and their oral bioavailability. While it is true that nutraceuticals are active in vitro, their in vivo activity is linked to these properties which, in the case of orientin, are still to be determined (as far as I could see).
It would probably be best if the first paragraph of the discussion was moved to the end, just before "Collectively, this study lays an important foundation for future investigations to confirm these findings in other experimental models of insulin resistance or T2D. This is in line with also checking the comparative or combination use of this flavone with other antidiabetic agents, since there were no synergistic effects observed in combining orientin and insulin to reverse hepatic insulin resistance in the current study". This way, the authors would first sum up their results and then present their conclusions in a more linear fashion.

I consider this communication to be worthy of publication on Molecules, with minor revisions necessary mostly for the sake of readability.
